# The Effect of Stimulation Protocols (GnRH Agonist vs. Antagonist) on the Activity of mTOR and Hippo Pathways of Ovarian Granulosa Cells and Its Potential Correlation with the Outcomes of In Vitro Fertilization: A Hypothesis

**DOI:** 10.3390/jcm11206131

**Published:** 2022-10-18

**Authors:** Michail Papapanou, Kalliopi Syristatidi, Maria Gazouli, Makarios Eleftheriades, Nikolaos Vlahos, Charalampos Siristatidis

**Affiliations:** 1Assisted Reproduction Unit, Second Department of Obstetrics and Gynecology, “Aretaieion” Hospital, Medical School, National and Kapodistrian University of Athens, 76 Vas. Sofias Av., 11528 Athens, Greece; 2Second Department of Obstetrics and Gynecology, “Aretaieion” Hospital, Medical School, National and Kapodistrian University of Athens, Vas. Sofias 76, 11528 Athens, Greece; 3Obstetrics, Gynecology and Reproductive Medicine Working Group, Society of Junior Doctors, 15123 Athens, Greece; 4School of Medicine, University of St. Andrews, North Haugh, St. Andrews KY16 9TF, UK; 5Laboratory of Biology, Medical School, National and Kapodistrian University of Athens, 176 Michalakopoulou Str., 11527 Athens, Greece

**Keywords:** mechanistic target of rapamycin, Hippo signaling pathway, ovarian stimulation protocol, GnRH agonist, GnRH antagonist, in vitro fertilization, assisted reproductive techniques

## Abstract

Controlled ovarian hyperstimulation (COH) is essential for the success of in vitro fertilization (IVF). Evidence showing the comparison of different COH protocols remains predominantly of low certainty and derives from unspecified infertile and highly heterogeneous populations. Thus, personalized approaches to examine the response of patients to the various COH protocols need to be investigated. Data from in vitro and animal studies have identified the mechanistic target of rapamycin (mTOR) and Hippo signaling pathways play a key role in follicular homeostasis and oocyte quality. To be specific, current data indicate the controlled activation of mTOR and the controlled inhibition of the Hippo pathway within the ovarian granulosa cells (GC). Both are reported to lead to a nurturing follicular microenvironment, increase oocyte quality, and potentially improve reproductive outcomes. As intracellular markers, phosphorylated/unphosphorylated levels of the pathways’ main downstream mediators could be included among the candidate “personalized” predictors of patients’ response to COH protocols and final IVF outcomes. Based on these hypotheses, we make a preliminary attempt to investigate their validity: We propose a prospective cohort study to compare the levels of certain phosphorylated/unphosphorylated components of the investigated pathways (mTOR, ribosomal protein S6 kinase beta-1 (p70S6K-1), yes-associated protein-1 (YAP-1), and transcriptional coactivator with PDZ-binding motif (TAZ)) within the follicular fluid-isolated GC between women undergoing gonadotropin-releasing hormone (GnRH) antagonist/“short” protocols and those receiving GnRH agonist/“long 21” protocols. A case-control design comparing these levels between women achieving pregnancy and those who did not is further planned. Additional analyses addressing the population’s expected heterogeneity are planned after the completion of the pilot phase, during which 100 participants undergoing IVF are intended to be recruited. At this stage, these hypotheses are solely based on in vitro/animal data, and thus, similar studies on humans in this respect are necessary for the investigation of their potential validity.

## 1. Introduction

### 1.1. Controlled Ovarian Hyperstimulation (COH) Protocols

Controlled ovarian hyperstimulation (COH) is a determining stage for the success of in vitro fertilization (IVF) as it leads to the necessary pooling of oocytes available for retrieval and subsequent creation of embryos for intrauterine transfer [1]. COH may be divided into three distinguishable phases: firstly, stimulation of multi-follicular development (for which the following have been used: human menopausal gonadotrophin (hMG), a purified (p-FSH) or highly purified follicle stimulating hormone (hp-FSH), a urinary product with follicle-stimulating hormone (u-FSH) and luteinizing hormone (LH) activity, and various recombinant FSH (rFSH) and LH (rFSH/rLH)); secondly, pituitary suppression towards the prevention of the LH surge and premature ovulation through the administration of gonadotropin-releasing hormone (GnRH) agonists or antagonists; and finally, the triggering of oocyte maturation and ovulation by using human chorionic gonadotropin (hCG) or GnRH agonists [2].

There is a continuing attempt to compare the efficacy and safety of the various COH protocols [2,3]. Their implementation, based on the administration of agonists of gonadotropin-releasing hormone (GnRH) (“long”), likely leads to comparable live birth rates (LBR) to those based on the GnRH antagonists (“short”). The latter reduces the probability of ovarian hyperstimulation syndrome (OHSS) in predicted normal or high responders [4,5,6]. The use of rFSH in COH in normal responders undergoing the “long” GnRH agonist protocols may also lead to lower fresh-cycle LBR compared to that of when hMG is used [5]. However, there are certain noteworthy limitations of the existing very low- to low-certainty evidence. Such include the lack of high-quality trials, the inclusion of unspecified populations of infertile couples in the vast majority of trials, and the insufficiency of data to allow separate estimations for poor and high responders or comparison of all possible combinations of GnRH agonists/antagonists with the gonadotrophin preparations and triggering agents [4,5]. The significant inconsistency between studied populations and interventions applies to most of the literature on assisted reproductive techniques (ART). Furthermore, it has been identified as one of the main research-related factors responsible for the plateauing of ART success rates [7,8]. Further potential reasons include flaws in the design, execution, and/or reporting of the current studies, as well as factors linked with issues related to clinical practice, such as the lack of the true efficacy of “add-on” treatments and the “industrialization” of IVF [7,8,9]. It therefore seems that the exploration and the proper use of markers that could provide clinicians with a more individualized picture of a patient’s probability of an adequate response to a specific COH protocol are, nowadays, crucial [10,11,12,13].

It could also be hypothesized that different categories of infertile patients undergoing IVF may respond differently to specific COH protocols, and consequently, this may lead to different outcomes [12]. The administered regimens affect the hormonal profile of each woman in a unique way, which is linked with her characteristics, such as demographic parameters or her personal unique history (e.g., different types of infertility such as ovulatory or tubal disorders, or endometriosis). In this respect, there is a need to investigate the intracellular pathways that may better reflect the effects of COH in combination with the individual profile of each patient separately. Two such pathways in ovarian granulosa cells (GC) that play an important role in follicular development are the phosphatidylinositide triphosphate kinase (PI3K)/Ak strain transforming (Akt) protein kinase/mechanistic target of rapamycin (mTOR) and Hippo/yes-associated protein-1 (YAP-1)—transcriptional coactivator with PDZ-binding motif (TAZ) [14].

### 1.2. The mTOR-Related Pathways in Ovarian GC

mTOR is a serine-threonine kinase that acts as the active catalytic subunit of two intracellular complexes, namely, the mTOR complex 1 and mTOR complex 2. A plethora of studies have demonstrated that mTOR may be activated by multiple signaling pathways within the ovarian GCs, including the PI3K/Akt and the cyclic adenosine monophosphate (cAMP)/cAMP-dependent protein kinase A (PKA)/mitogen-activated protein kinase (MAPK)/extracellular signal-regulated kinases (ERK) [14,15,16,17,18,19]. These exhibit dynamic interactions with one another and can be activated by various factors, including the FSH, the epidermal growth factor (EGF), the vascular epithelial growth factor (VEGF), and the insulin or insulin growth factor-1 (IGF-I) [14,16,19,20,21]. Ribosomal protein S6 kinase beta-1 (p70S6K-1) and eukaryotic translation initiation factor 4E (eIF4E) are the main downstream effectors of the mTOR pathway [15,22,23].

Accumulating evidence from in vitro studies or animal models has demonstrated that the stimulation of mTOR and its downstream effectors in ovarian GCs may orchestrate the processes of quiescence, activation, and survival of primordial follicles; induce proliferation and differentiation of GCs and meiotic maturation of oocytes [24,25,26]; and suppress the phenomena of autophagy and apoptosis [27,28,29,30]. The activation of mTOR via the described pathways seems to be an essential mediator for the bidirectional communication between GCs and oocytes. These interactions lead to a nurturing microenvironment for follicular growth, which in turn empowers the oocyte’s developmental potential [31,32,33,34]. A better oocyte developmental competence is associated with an increased ability of the oocyte to complete meiosis and, subsequently, undergo fertilization, implantation, embryogenesis, and term development [31,33]. In contrast, the overactivation of mTOR-related pathways in in vitro or animal studies has been associated with ovarian aging and the pathogenesis of premature ovarian insufficiency due to massive maturation of primordial follicles [35,36], polycystic ovary syndrome (PCOS) [37,38], endometriosis [39], and ovarian cancers [40,41,42]. In the same context, its inhibition may lead to an increase in ovarian reserve and lifespan [43,44] and the mitigation of OHSS symptom severity [45].

### 1.3. The Hippo Pathway in Ovarian GC

The Hippo pathway is mediated by the phosphorylation of the mammalian sterile 20-related kinases 1 and 2 (Mst1/2), which activate the large tumor suppressor kinases 1 and 2 (Lats1/2). The latter, phosphorylate the translation activators YAP-1 and TAZ, which thus remain outside the nucleus and are degraded in the cytoplasm [46,47,48]. Hippo is inhibited during ovarian stimulation through cytoskeletal reorganization, extracellular matrix stiffness, ovarian fragmentation, and the activation of G protein-coupled receptors, integrins, or growth factors [46,49]. Thus, the unphosphorylated YAP-1 and TAZ activators freely enter the nucleus, form complexes with TEA domain family member transcription factors (TEADs), and further stimulate the transcription of proliferation-related and growth factor genes [47,48,50].

Like the mTOR-related pathways, the Hippo pathway plays a key role in the control of follicular growth and homeostasis. The Hippo pathway is reported to regulate the balance between GC proliferation (which dominates before ovulation when Hippo is predominantly inactive), steroidogenesis, and differentiation of the cells into a luteal phase (processes that dominate after ovulation when Hippo is active) [14,47]. The Hippo pathway effectors interact with the EGF-like signaling [51], as well as the components of the aforementioned intracellular signaling involving mTOR [12,14,52]. Excessive inhibition of the Hippo pathway and unrestrained inflow of the chief final mediators YAP-1 and TAZ into the nucleus have been involved in the development of PCOS [46,53], endometriosis [54,55], and ovarian tumorigenesis [56]. On the other hand, in vitro fragmentation and controlled inhibition of Hippo (combined with administration of PI3K/Akt/mTOR activators) can promote the growth of preantral follicles in patients with primary ovarian insufficiency [57,58,59]. According to a recent randomized controlled trial, the fragmentation of one ovary for follicular activation compared to no intervention in women with poor ovarian response resulted in an increase of the antral follicle count (AFC), an 18.8% reduction in the phosphorylated YAP-1/unphosphorylated YAP ratio (indicating Hippo pathway inhibition), but this had no impact on the IVF outcomes [60].

Taking all the above into consideration, the controlled activation (i.e., phosphorylation) of mTOR and its related downstream effectors (e.g., p70S6K-1) and controlled inhibition of Hippo (i.e., reduction of phosphorylated YAP-1, TAZ) are of potential benefit for the ovarian GC-oocyte interactions, oocyte quality, and reproductive outcomes. This hypothesis remains to be reinforced or disproved in humans.

## 2. Objectives

In this paper, we aim to express the hypothesis that the activity of both the mTOR (the mechanistic target of rapamycin) and Hippo pathways within the GCs isolated by the follicular fluid of women undergoing IVF may be affected by the type of protocol used for COH; these differences in activity may also influence the outcomes of the IVF cycle. As discussed above, these pathways orchestrate the processes of proliferation, differentiation, and apoptosis of GCs. Such processes are directly related to the development and maturation of the follicles, the quality of the oocytes, and potentially, the outcomes of the IVF cycle. Data to form this hypothesis currently derive from in vitro/animal studies; thus, this study design involving humans is original in the literature.

To initially test the potential validity of this hypothesis, we propose the measurement of the activity of the two pathways through the levels of the phosphorylated/unphosphorylated forms of specific proteins: i. mTOR, p70S6K-1 to reflect the downstream activation of mTOR-related pathways and ii. YAP-1, TAZ to reflect the activation of Hippo. The phosphorylated forms of mTOR and p70S6K-1 represent the increased activation of the mTOR-related signaling pathways, whereas the decreased phosphorylated and elevated unphosphorylated levels of YAP-1 and TAZ signify the Hippo inhibition, and therefore intensified the signals for GC proliferation. Levels of these forms will be compared between GCs isolated from the follicular fluids of women receiving GnRH agonists and those receiving GnRH antagonists for pituitary suppression. This first stage of the study will be conducted prospectively.

We further aim to compare these measurements between women achieving positive IVF outcomes and those who did not. In this way, we will attempt to examine whether the potential differences in the activity of the two pathways could be correlated with the final (primary and/or secondary) outcomes of the IVF cycle. This second stage will be conducted through a case-control design. The study is aimed to be divided into an initial pilot phase, during which the feasibility of the design will be assessed, and a final phase, during which a growing registry is to be created where additional subgroup analyses (addressing the inconsistency of the population) are to be conducted.

The upper goal of this research effort is to investigate whether measuring the activity of these intracellular pathways can somehow predict the ovarian response to COH protocols and/or final IVF outcomes by reflecting a more personalized state of the microenvironment of the ovarian GCs. The need for immediate study and further knowledge of these paths arises from the growing expectations of infertile couples to increase the chance of successful outcomes at the lowest possible financial cost while minimizing clinical procedures (reduction of the number of IVF attempts) and/or the probability of unwanted outcomes (e.g., miscarriage, ectopic pregnancy, and OHSS).

## 3. Materials and Methods

### 3.1. Study Design

Comparison of the phosphorylated/unphosphorylated levels of the pathways’ components following the administration of GnRH antagonist/“short” protocols versus GnRH agonist/“long 21” protocols will be conducted through a prospective-cohort design. A comparison of levels between women achieving certain predefined outcomes within the IVF cycle and those not reaching these outcomes will be carried out through a case-control design. The pilot phase of the study, during which 100 women undergoing IVF are planned to be recruited, will be conducted in the Second Department of Obstetrics and Gynecology of “Aretaieion Hospital”, Athens, Greece, a tertiary academic center within a university IVF unit (Figure 1). Institutional review board approval will be received, and informed consent will be obtained from all participants. Following the potential success of the pilot phase, more public/private study centers are to be added towards the increased recruitment of the study subjects.

### 3.2. Eligibility Criteria and Data Collection

All women aged 30–40 years old undergoing IVF due to female/male factor or unexplained infertility will be considered eligible for the study. Advanced age has been identified as a negative factor for IVF success rates. The limit is around 40 years of age. After this age, a steep decline in pregnancy and delivery rates are observed [61,62]. As noted above, and based on the currently available evidence, GnRH agonist and GnRH antagonist COH protocols lead to comparable live birth rates [4]. For all these reasons, we chose to include women in the age range of 30 to 40. Two protocols of i. GnRH agonists and ii. GnRH antagonists for pituitary suppression, following stimulation with r-hFSH, will be administered to participants.

For the GnRH agonist protocol, Triptorelin [Gonapeptyl, 3.75 mg (Ferring Pharmaceutical Hellas A.Ε.) or Arvekap, 3.75 mg (Ipsen, EPE)] will be administered subcutaneously daily during the midluteal phase of the menstrual cycle.

For the GnRH antagonist protocol, ovarian stimulation will begin on the second day of the cycle, and the antagonist, either Cetrorelix (Merck Serono Europe Limited, Middlesex, UK) or Orgalutran (Merck Sharp & Dohme Limited, Hoddesdon, UK), will be initiated as soon as the leading follicle reaches a diameter of 14 mm. Once pituitary downregulation and ovarian suppression is achieved, ovarian stimulation with exogenous gonadotropins will start, while GnRH agonist administration will continue concomitantly until the day of hCG administration. Recombinant FSH in the form of either follitropin alpha (Gonal-F; Merck Serono Europe Ltd, London, UK) or follitropin beta (Puregon; Merck Sharp & Dohme Ltd., Hertfordshire, UK) will be administered subcutaneously.

Women outside the age span mentioned above or those undergoing natural or mild IVF protocols will be excluded from the study. The following patient data will be collected: age; BMI; type of infertility (i.e., male, female, or unexplained); relevant comorbidities associated with infertility (e.g., ovulatory disorders such as PCOS, tubal factors, or endometriosis); data from previous IVF cycles; levels of anti-Mullerian hormone (AMH); FSH; LH; prolactin; AFC; estradiol (E2) and serum progesterone in the early follicular phase; and data on the IVF cycle of the pathways’ measurements, such as the number and size of follicles at the time of triggering, and the number of aspirated follicles and oocytes retrieved. Successful or unwanted outcomes of the IVF cycle of the pathways’ measurements will also be recorded. Data will be collected within a secure online platform, further ensuring patient anonymity. Access to the platform will be granted strictly to members involved in the collection, analysis, or interpretation of data.

### 3.3. The Biological Samples

After the separation of the oocytes from the follicular fluid by the embryologists, the follicular fluid will be collected in a sterile test tube and transferred to the Laboratory of Biology of the Medical School of National and Kapodistrian University of Athens, Athens, Greece. The GC will be purified from the follicular fluid using the red blood cell lysing buffer (RLB) method, as described previously [63]. The measurement of the pathways’ activity will be performed on the isolated GC from the collected and retained follicular fluids using special ELISA kits (e.g., Phospho-mTOR (S2448) and Total mTOR ELISA Kit (ab279869), Human/Phospho-P70S6K (T421/S424) ELISA Pharma (SBRS1919), PathScan^®^ Phospho-YAP (Ser127) Sandwich ELISA Kit #48804, Human TAZ ELISA Kit) to measure phosphorylated forms of specific pathway proteins (mTOR, p70S6K-1 for mTOR-related activation and YAP-1 and TAZ for Hippo).

### 3.4. Intended Comparisons and Outcomes

The phosphorylated/unphosphorylated levels of the four downstream effectors of the investigated pathways within the isolated GCs will be compared between women receiving a GnRH agonist protocol and those being administered a GnRH antagonist protocol. Furthermore, a comparison of the activity of these two pathways will be made concerning certain desired or unwanted outcomes of the IVF cycle, during which the pathways’ measurements took place. Live birth after the cycle of measurements will serve as the primary outcome of the case-control study. Clinical pregnancy (defined as the presence of a gestational sac on ultrasound scan at 6–7 weeks of gestation), implantation rate (defined as the number of sacs detected divided by the number of embryo transfers), ongoing pregnancy (i.e., presence of a positive fetal heartbeat on ultrasound from a gestational age of 10 weeks onwards), miscarriage (i.e., loss of pregnancy before 22 completed weeks of gestational age), ectopic pregnancy, and OHSS will be the secondary outcomes of the study.

### 3.5. Data Analysis

Categorical variables will be presented as frequencies and percentages, whereas numerical variables will be presented as mean (standard deviation) or median (interquartile range) based on the normality of the data. A normality check will be performed by using the Shapiro–Wilk test. Independent samples *t*-test or Mann–Whitney U test (as per normality or not of data) will be utilized for comparisons regarding numerical variables, whereas chi-square or Fisher’s exact (if applicable) tests will be used for categorical variables. Following the pilot phase (during which the following analyses would be underpowered), further analyses are being planned. Comparisons of the pathways’ activity will be made by patients’ age (<35 vs. >35 years of age), by category of infertility (male, female, or unexplained), and by the female infertility-related factor (e.g., PCOS and endometriosis). Concerning the IVF outcomes, multivariable logistic regression models are further being planned.

## 4. Conclusions

Based on data mostly derived from animal models or in vitro studies, it seems that controlled activation of mTOR, its related downstream effectors, and controlled inhibition of the Hippo pathway within the ovarian GCs may lead to a nurturing follicular microenvironment, reinforce the beneficial ovarian GC-oocyte interactions, and therefore positively influence oocyte quality and reproductive outcomes. Phosphorylated/unphosphorylated levels of the pathways’ main downstream mediators may be included among the intracellular biomarker candidates that feasibly provide a more “personalized” prediction of patients’ response to certain COH protocols and final IVF outcomes. Only at a very preliminary stage of this suggested effort, our study group proposes a prospective cohort study to compare the levels of certain phosphorylated/unphosphorylated components of the investigated pathways (mTOR, p70S6K-1, YAP-1, and TAZ) between GC isolated from the follicular fluid of women undergoing GnRH antagonist/“short” protocols and the ones from women receiving GnRH agonist/“long 21” protocols. After recording the desired/unwanted primary and secondary outcomes of the IVF cycle, a case-control design comparing these levels between women achieving the predefined endpoints and those who did not is further planned. Following the successful completion of the pilot phase, and given higher expected recruitment at later stages of the project, further analyses stratifying women by age groups and/or different etiological factors of infertility are planned to address the heterogeneity of the population. Notably, many studies in this direction are needed before any conclusions on the validity of the stated hypothesis can be drawn.

## Figures and Tables

**Figure 1 jcm-11-06131-f001:**
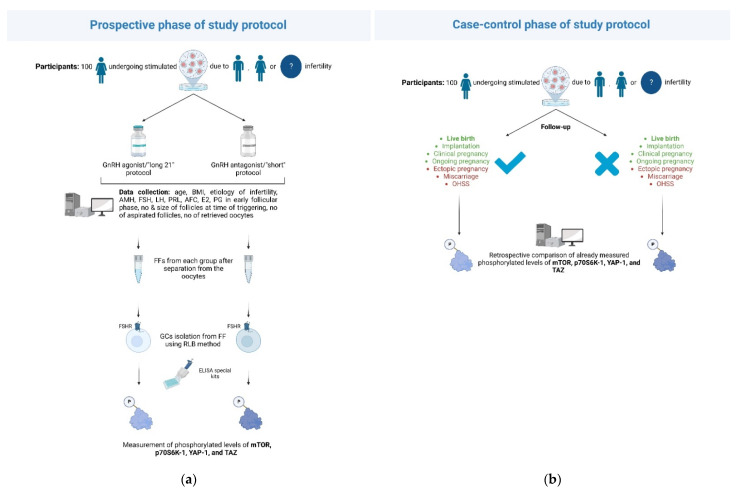
Illustration of the prospective and case-control phases of the planned study. Eligible participants will be women aged 30–40 undergoing stimulated IVF cycle (i.e., not undergoing natural or mild IVF protocols) due to infertility of male, female factor, or unexplained etiology. The number of participants (100) refers to the targeted sample of the pilot phase. Following the successful completion of this phase, more study centers and subjects are intended to be included. (**a**) Prospective phase of the designed study. Levels of certain phosphorylated components of the investigated mTOR and Hippo pathways will be compared between women receiving GnRH agonist/“long 21”protocols and those receiving GnRH antagonist/”short” protocols. These levels will be measured via special ELISA kits by using follicular fluid-isolated ovarian GCs from each group. (**b**) Case-control phase of the planned study. Already measured levels of certain phosphorylated components of the investigated mTOR and Hippo pathways will be retrospectively compared between women achieving predefined (positive/negative) IVF endpoints and those not. Both schemes were created with Biorender.com (www.biorender.com, 19 September 2022). A license for the use of BioRender icons appearing in both schemes and the publication has been granted. Both schemes are original and were created by the authors. ***Abbreviations:*** GnRH, gonadotropin-releasing hormone; BMI, body mass index; AMH, anti-Mullerian hormone; FSH, follicle stimulating hormone; LH, luteinizing hormone; PRL, prolactin; AFC, antral follicle count; E2, estradiol; PG, progesterone; no, number; FF, follicular fluid; GC, granulosa cell; RLB, red blood cell lysing buffer; ELISA, enzyme-linked immunosorbent assay; mTOR, the mechanistic target of rapamycin; p70S6K-1, ribosomal protein S6 kinase beta-1; YAP-1, yes-associated protein-1; TAZ, transcriptional coactivator with PDZ-binding motif; OHSS, ovarian hyperstimulation syndrome.

## Data Availability

Not applicable.

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
