# Peer review of "The Effect of Stimulation Protocols (GnRH Agonist vs. Antagonist) on the Activity of mTOR and Hippo Pathways of Ovarian Granulosa Cells and Its Potential Correlation with the Outcomes of In Vitro Fertilization: A Hypothesis"

_jcm, 2022, doi:10.3390/jcm11206131_

Round 1

Reviewer 1 Report (New Reviewer)

1. What is the age effection in this COH protocol?  Could authors disciss more about the effect on 30~40 years old?

2. Results were little complex in this manuscript. Is it possible to offer the figure to express the final results or conclusion? 

Author Response

We would like to thank you for the effort and valuable time spent in reviewing our manuscript (ID: jcm-1952799) entitled “The effect of stimulation protocols (GnRH agonist vs. antagonist) on the activity of mTOR and Hippo pathways of ovarian granulosa cells and its potential correlation with the outcomes of in vitro fertilization: a hypothesis”. We have taken into consideration all comments and, herein, provide a detailed point-by-point response. All changes have been highlighted using the “track changes” function of Microsoft Word.

  1. What is the age effection in this COH protocol? Could authors disciss more about the effect on 30~40 years old?.

We sincerely thank the reviewer for the time spent in reviewing our manuscript and the valuable comments.

Most ART treatments take place in women aged between 30 and 39 (ART fact sheet +10, ESHRE). Advanced age has been identified as a negative factor for IVF success rates. The limit is around 40 years of age, where after this, a steep decline of pregnancy and delivery rates is observed (https://doi.org/10.1093/hropen/hoaa032, https://doi.org/10.1093/hropen/hoab026). It should be further noted that, based on currently available evidence, GnRH agonist and GnRH antagonist COH protocols lead to comparable live birth rates (10.1002/14651858.CD001750.pub4). For all these reasons, we chose to include women at the age range of 30-40. In order to address a potential heterogeneity within this range, we further plan a subgroup analysis by maternal age (<35 years old vs. > 35 years old). We have highlighted the above points within the text, supporting them with some additional references, as suggested.

  1. Results were little complex in this manuscript. Is it possible to offer the figure to express the final results or conclusion?.

We thank the reviewer for the suggestion. We are at the preliminary stage of waiting for the ethical approval in order for recruitment of patients to begin; that is why no results or real data are still presented. In this manuscript, we aimed to present a concise, yet up-to-date, mini review of in vitro, animal, or preclinical human data creating the rationale for the study, forming and presenting the research hypothesis and objectives of the study (both short-term and long-term), and describe its planned design (i.e., research protocol). For this reason, we have submitted the manuscript as a “communication” paper rather than a “clinical study”. To address the complexity of the text, as suggested, we have modified many parts of the manuscript (changes are highlighted throughout the manuscript). We have also created a figure summarizing the stages of the planned study (Figure 1a, 1b).

Reviewer 2 Report (Previous Reviewer 2)

This study investigated the effect of the activation of mTOR and Hippo signaing
pathways on the difference in patients' response to COH protocols, and
furthermore, on oocyte quality and pregnancy outcome using granulosa cells
obtained from follicular fluid during IVF. The idea of this study is novel
and very interesting,considering that the activation of mTOR and Hippo signaling
pathways has been shown to play an important role in the regulation of ovarian
microenvironment for follicledevelopment.
Unfortunately, however, it is very difficult to evaluate
any relationships between the results and the purpose of this study,
because there is no part of any evidences to prove such results in this
manuscript. In addition, it was described in complex sentences that were
difficult for readers to understand this article. Therefore, this article
is needed to be corrected and rewritten some parts.

Author Response

We would like to thank you for the effort and valuable time spent in reviewing our manuscript (ID: jcm-1952799) entitled “The effect of stimulation protocols (GnRH agonist vs. antagonist) on the activity of mTOR and Hippo pathways of ovarian granulosa cells and its potential correlation with the outcomes of in vitro fertilization: a hypothesis”. We have taken into consideration all comments and, herein, provide a detailed point-by-point response. All changes have been highlighted using the “track changes” function of Microsoft Word.

This study investigated the effect of the activation of mTOR and Hippo signaing
pathways on the difference in patients' response to COH protocols, and
furthermore, on oocyte quality and pregnancy outcome using granulosa cells
obtained from follicular fluid during IVF. The idea of this study is novel
and very interesting,considering that the activation of mTOR and Hippo signaling
pathways has been shown to play an important role in the regulation of ovarian
microenvironment for follicledevelopment.

We sincerely thank the reviewer for the time spent in reviewing our manuscript and the valuable comments.

Unfortunately, however, it is very difficult to evaluate
any relationships between the results and the purpose of this study,
because there is no part of any evidences to prove such results in this
manuscript. In addition, it was described in complex sentences that were
difficult for readers to understand this article. Therefore, this article
is needed to be corrected and rewritten some parts.

We thank the reviewer for the comment. We are at the preliminary stage of waiting for the ethical approval in order for official recruitment of patients to begin; that is why no results or real data are presented. Herein, we aimed to present a concise, yet up-to-date, mini review of in vitro, animal, or preclinical human data creating the rationale for the study, forming and presenting the research hypothesis and objectives of the study (both short-term and long-term), and describe its planned design (i.e., research protocol). For this reason, we have chosen to submit the manuscript as a “communication” paper rather than a “clinical study”. It could be considered a letter of intent/protocol and a mini review of the relevant literature. To address the complexity of the text, as suggested, we have corrected many parts of the manuscript (changes are highlighted throughout the manuscript).

Finally, we would like to inform you that both schemes (Figure 1a and 1b) were created with Biorender.com (www.biorender.com). License for use of Bio-Render icons appearing in both schemes and publication in the Journal of Clinical Medicine has been granted (Agreement Numbers: TG24I6Z5EX and OK24I6YVNR). Both schemes are original and were created by the authors.

All in all, we hope that with these changes our manuscript is felt appropriate to be published in the Journal of Clinical Medicine. We always remain at your disposal should you require any further information on our manuscript and are looking forward to receiving your response.

Sincerely yours,

Michail Papapanou, MD

Prof. Charalampos Siristatidis, MD, PhD

Second Department of Obstetrics and Gynaecology, “Aretaieion” University Hospital, Medical School, National and Kapodistrian University of Athens

This manuscript is a resubmission of an earlier submission. The following is a list of the peer review reports and author responses from that submission.

Round 1

Reviewer 1 Report

The aim of the present study results interesting considering the novelty of the field and the importance of the topic. 

The clinical hypothesis of study is  supported by animal evidences but in my opinion authors should improve the materials and methods section. It would be better describing more in detail the process of granulosa cells isolation from follicular fluid and the COH stimulation in all of its phases (i.e gonadotropins administered for follicular growth, final oocyte maturation. etc...)   

Reviewer 2 Report

This study investigated the effect of the activation of mTOR and Hippo signaling pathways on the difference in patients’ response to COH protocols, and furthermore, on oocyte quality and pregnancy outcome using granulose cells obtained from follicular fluid during IVF. The idea of this study is novel and very interesting, considering that activation of mTOR and Hippo signaling pathways has been shown to play an important role in the regulation of ovarian microenvironment for follicle development. Unfortunately, however, it is very difficult to evaluate the results for the purpose of the study because there is no part of the results to prove such results in this manuscript. In addition, it is described in complex sentences that are difficult for readers to understand throughout this manuscript, so it needs to be corrected.

Reviewer 3 Report

This is a strange manuscript only describing the design of a forthcoming study. No results or real data are presented. It´s more or less a letter of intent.